# Design and Simulation Studies of Hybrid Power Systems Based on Photovoltaic, Wind, Electrolyzer, and PEM Fuel Cells

**Hussein A.Z. AL-bonsrulah** [1,*], **Mohammed J. Alshukri** [2], **Lama M. Mikhaeel** [3], **Noor N. AL-sawaf** [4], **Kefif Nesrine** [5], **M.V. Reddy** [6,*] and **Karim Zaghib** [7]

1   Department of Energy Engineering, Sharif University of Technology, Azadi Avenue, Tehran 14588-89694, Iran
2   Department of Mechanical Engineering, Faculty of Engineering, Kufa University, Najaf 54002, Iraq; mohammedj.alshukry@uokufa.edu.iq
3   Department of Industrial Automation and Control Engineering, Tartous University, Tartus C5335, Syria; lama.mikhaeel.93@gmail.com
4   Department of Electrical Engineering, University of Mosul, Mosul 41001, Iraq; noornawaf81@gmail.com
5   Department of Power and Control engineering, Institute of Electrical and Electronic Engineering (IGEE), University of M'hamed Bougara, Boumildas 35000, Algeria; kefifn92@gmail.com
6   Centre of Excellence in Transportation Electrification and Energy Storage (CETEES), Institute of Research Hydro-Québec, 1806, Lionel-Boulet Blvd., Varennes, QC J3X 1S1, Canada
7   Department of Mining and Materials Engineering, McGill University, Wong Building, 3610 University Street, Montreal, QC H3A OC5, Canada; karim.zaghib@mcgill.ca
*   Correspondence: huseenabd541@gmail.com (H.A.Z.A.-b.); reddymvvr@gmail.com (M.V.R.)

**Abstract:** In recent years, the need to reduce environmental impacts and increase flexibility in the energy sector has led to increased penetration of renewable energy sources and the shift from concentrated to decentralized generation. A fuel cell is an instrument that produces electricity by chemical reaction. Fuel cells are a promising technology for ultimate energy conversion and energy generation. We see that this system is integrated, where we find that the wind and photovoltaic energy system is complementary between them, because not all days are sunny, windy, or night, so we see that this system has higher reliability to provide continuous generation. At low load hours, PV and electrolysis units produce extra power. After being compressed, hydrogen is stored in tanks. The purpose of this study is to separate the Bahr AL-Najaf Area from the main power grid and make it an independent network by itself. The PEM fuel cells were analyzed and designed, and it were found that one layer is equal to 570.96 Watt at 0.61 volts and 1.04 A/Cm². The number of layers in one stack is designed to be equal to 13 layers, so that the total power of one stack is equal to 7422.48 Watt. That is, the number of stacks required to generate the required energy from the fuel cells is equal to 203 stk. This study provided an analysis of the hybrid system to cover the electricity demand in the Bahr AL-Najaf region of 1.5 MW, the attained hybrid power system TNPC cost was about 9,573,208 USD, whereas the capital cost and energy cost (COE) were about 7,750,000 USD and 0.169 USD/kWh respectively, for one year.

**Keywords:** hybrid power system; PEM fuel cell; renewable energy; photovoltaic (PV); wind turbine; economic technical

## 1. Introduction

The most widely used alternative energy sources are wind and solar. Wind power is being regarded as the main source of renewable power generating capacity in Europe, the United States, and China in 2019, as per the 'reports of Renewables Global Status (REN21) (2020)'. A total of 60 GW of renewable wind power capacity was installed globally, bringing the total to 651 GW. Many companies and private corporations are switching to this type of power source because of its reliability with lower operating costs. Besides that, several major investors are drawn to it because of the consistent profits it generates [1]. In markets such as Japan, India, China, and the USA, solar PV was the main producer

of renewable energy power in 2019. Approximately 115 GW of solar PV capacity went into service internationally (on/off-grid), nearly doubling the current total capacity of 627 GW [2]. Multi-electrical power production sources are referred to as a hybrid energy system (HES). It is a device that incorporates various renewable energy sources, either alone or in combination with traditional energy sources, including diesel generators (DG). Such sources have various loads that are attached to components of storage to remunerate for the discontinuous renewable energy sources' nature and increase total energy quality [3]. The biggest advantage of the hybrid energy system network is that it is self-adequate in a variety of climates and it does not depend on one source. HES is operated independently as a micro-grid or integrated into the electrical grid. Numerous strategies of management for a separate HES mode have been suggested in [4], and the authors examined three HES strategies of management in a faraway region with PV/wind/PEM-FC. Wind and PV have been used as original sources, while PEM-FC was used as a backup or secondary source. The system aimed to thicken the FC membrane and guarantee that energy was delivered from the sources to the load. A two-level controller was suggested in [5] to increase the power equilibrium, reliability of the system, and solve the limitations of traditional methods in HES energy management (PV, FC, and battery storage). A connected controller of MPPT-droop dual-configuration is implemented at the operational level, while a corresponding economic independence technique for shared system total power between the FC the battery pack is employed at the network level. A system of Fuzzy Logic Control (FLC) within the system of energy control has been proposed in [6] for a PV/hydro/WT/FC/battery hybrid energy system to decrease peak demand and eliminate system expense. The used FLC system has employed a program logic control system. In [7], two innovative power management techniques according to the optimization of mine blast for FC, SC, and battery hybrid systems and dependent on the slap swarm algorithm (SSA) are suggested, considering the response of demand and the management mode's storage of hydrogen system to satisfy the load demand. The proposed solutions were compared to current approaches, such as maximization of external energy [8], Fuzzy logic control [9,10], the state machine, and classical proportional-integral control using the consumption of hydrogen fuel and performance key points. In contrast, a consumer-side power control system for combining with the estimating scheme has been introduced in [11] for HES linked to the electric grid [12,13]. The system of pre-processed metering was incorporated in this design, along with a control mechanism that worked based on the consumer's preferences. On a monthly timescale, this model helped to restrict the consumption of renewable energies. The researchers of [14] suggested a system of PV coupled with the electrical network for supplying a load of DC by that kind of electric grid but without splitting any excess power which can be pumped into the electricity network with a significant effect. The battery voltage was regulated using signals that indicated the charging process or if the battery was being discharged more deeply [15] to choose the bi-directional converter's operating mode (boost or buck). The PV array and FC were used as the primary and secondary sources, accordingly, while the SC and BSS were used as energy storage elements to satisfy load demand. They contrasted their control method to a traditional control scheme dependent on the PI controller in a variety of time models. Experimental work is still needed. Therefore, in [16], a PV/wind/battery micro-grid HES with a two-layer Energy Management Strategy was studied, in which they used convex modeling in the upper EMS layer to predict PV wind power production and load demand. The bottom EMS layer was devoted to the power distribution, which required to be provided through various sources of an HES employing a predictable model and trying to roll out the horizon control strategy. The key disadvantages of the aforementioned energy management strategy for the various HES configurations are slower converge speed, particularly for the system that requires an optimized objective function [17], and others are dependent on process variables and suffering from a voltage stability challenge throughout varying power reference. Analytical modeling has been carried out in [18] to investigate the influence of many parameters, such as the porosity, unit cell aspect ratio,

fiber radius, and molar concentration, on the transverse permeability of the gas diffusion layer (GDL). The fibrous porous media (porosity and fiber radius), the zeta potential, and the electrolyte solution's physical properties have been clearly stated in the suggested modeling. A comparison was made between the results obtained from the proposed modeling and the results of a number of previous literature studies, and it was found that there is a great relative match between these results. To measure the efficient electrolyte diffusivity in porous media while taking into account the influence of electrical double layer (EDL), fractal modeling of the porous media's fractal theory and capillary model have been suggested in [19]. The proposed modeling specifically addresses the electro-kinetic parameters as well as the porous media's microstructural parameters. To verify the validity of the results, a comparison was made between the modeling results and data for the results of an experimental study. It was found that there is an acceptable match between these results. In this paper, an investigative study on hybrid energy systems of PV, wind, electrolyzer, and PEM fuel cells was conducted in the Bahr Al-Najaf region. Bahr Al-Najaf region is an area located exactly in the Al-Nour region in southern Iraq. This region is characterized by a hot, dry climate in the summer, so the total demand for electricity consumed in this region increases. Also, it is suffering from the problem of the difficulty of providing electricity in an accessible way. The reason for this is because the total dependence in the supply of electric power is to extend national high-voltage lines that transmit electrical energy over long distances. Therefore, the losses of electrical energy across these long lines are relatively high, especially at the peak demand on hot summer days. The process of making the system hybrid was undertaken because wind and solar energy cannot be available at all times, so they are combined to enhance the power production from the system. The wind and PV energy systems are complementary because not all days are sunny or windy, or for use at night, so this system has higher reliability to provide continuous generation [20]. In the studied HES, the energy resulting from wind turbines or photovoltaic arrays can be used to produce hydrogen using an electrolyzer in the fuel cell which works in parallel with the other systems [21]. In addition to the HOMER simulation, a mathematical model of PEMFC was proposed as well as numerical method simulation, which is used for the purpose of analyzing the performance of the suggested hybrid energy system. A dynamic simulation model was developed that explains the design and analysis of this system and the monitoring and analysis of system performance for the solar system, and the output constant voltage is evaluated within the AC/DC exchanger and connected with the other components of the system [22].

## 2. Mathematical Model of PEM and Numerical Method

### 2.1. Equations of the Model System

2.1.1. Channels from Which Gas Flows

By solving steady-state gas flow field equations, "continuity equations", the gas flow field for the fuel cell channels is found:

$$\nabla \cdot (\rho \vec{V}) = 0 \tag{1}$$

where $\rho$ is density (in kg/m$^3$) and $\vec{V}$ is velocity vector (m/s) in x-, y-, and z-direction, respectively.

In addition, the momentum equation was written as:

$$\nabla \cdot (\rho \vec{V} \times \vec{V} - \mu \nabla \vec{V}) = -\nabla \left( P + \frac{2\mu \nabla \vec{V}}{3} \right) + \nabla \cdot \left( \mu \left( \nabla \vec{V} \right)^T \right) \tag{2}$$

where, $\mu$ is viscosity (kg/(m·s)) and P is pressure (in Pa).

According to the description of the difference of mass flow, mass balance through convection, and diffusion, the mass transfer equation in the steady-state can be written as:

$$\nabla \cdot \left( -\rho \; \varphi_i \sum_{j=1}^{j=N} \sigma_{ij} \frac{M}{M_j} \left[ \nabla \varphi_j + \varphi_j \frac{\nabla M}{M_j} \right] + \rho \; \varphi_i \cdot \vec{V} \right) = 0 \tag{3}$$

where T is the temperature (in K), $\varphi$ is mass fractions, M is the gas molecular weight (in kg/mole), $\sigma$ is the diffusion coefficient (in m$^2$/s), and the subscript letters i and j refer to hydrogen in anode or oxygen in cathode and the water vapor in both sides, respectively.

Based on both pressure and temperature, it is possible to determine Maxwell–Stefan diffusion coefficients from the kinetic theory equation [23]:

$$\sigma_{ij} = \left( \frac{M_i + M_j}{M_i M_j} \right)^{\frac{1}{2}} \frac{T^{\frac{7}{4}}}{P \left( \left( \sum_r W_{ri} \right)^{\frac{1}{3}} + \left( \sum_r W_{rj} \right)^{\frac{1}{3}} \right)^2} \times 10^{-3} \tag{4}$$

In this equation, the units of binary diffusion coefficient and pressure are cm$^2$/s and atm, respectively. The values for $\sum W_{ri}$ are provided in [23].

Moreover, the temperature field can be found by solving the following heat energy equation:

$$\nabla \cdot \left( \rho C_p T \vec{V} - k \nabla T \right) = 0 \tag{5}$$

where, k is thermal conductivity of gas (in W/(m·K)) and $C_p$ is specific heat capacity (in J/(kg·K)).

### 2.1.2. Channels from Which Gas Flows

The transportation model for the layer in which the gas is spread is executed and the continuity equation for these layers was obtained as:

$$\nabla \cdot \left( \rho \varepsilon \; \vec{V} \right) = 0 \tag{6}$$

where $\varepsilon$ is porosity. By reducing the momentum equation to Darcy's law, it can be obtained by:

$$\vec{V} = \frac{K_p}{\mu} \nabla P \tag{7}$$

where $K_p$ is the hydraulic permeability (in m$^2$).

For porous media, the equation of mass transportation can be obtained as:

$$\nabla \cdot \left( -\rho \varepsilon \; \varphi_i \sum_{j=1}^{j=N} \sigma_{ij} \frac{M}{M_j} \left[ \nabla \varphi_j + \varphi_j \frac{\nabla M}{M_j} \right] + \rho \varepsilon \; \varphi_i \cdot \vec{V} \right) = 0 \tag{8}$$

Employing the Bruggemann correction equation, it can correlate the diffusivities to estimate the porous media geometric constraints as:

$$\sigma_{ij}{}^{eff} = \sigma_{ij} \times \varepsilon^{3/2} \tag{9}$$

In addition, the heat transfer for the layer of gas diffusion can be determined as:

$$\nabla \cdot \left( \rho \varepsilon C_p T \vec{V} - k_{eff} \varepsilon \nabla T \right) = \varepsilon \beta (T_{solid} - T) \tag{10}$$

where β is a modified "convective heat transfer coefficient" which relates the porous medium's specific area (in $m^2/m^3$) with the heat transfer by convection (in $W/m^2$) [24]. Therefore, its approved unit is $W/m^3$.

In the diffusion layers, the potential distribution is subjected to:

$$\nabla \cdot (\lambda_e \nabla \varphi) = 0 \tag{11}$$

where $\lambda_e$ represents the electronic conductivity of the electrode (in S/m).

### 2.1.3. Layers of Catalyst

The layer of catalyst is considered as a thin face, due to the very small thickness. The sources for the reactors and terminology are accomplished and the source terminology is actually performed in the porous medium, especially the last grid cell on the side of the cathode. The sink term for the oxygen and hydrogen can be specified as:

$$S_{O_2} = -\frac{M_{O_2}}{4F} i_c \tag{12}$$

$$S_{H_2} = -\frac{M_{H_2}}{2F} i_a \tag{13}$$

where (F = 96,487 (C/mol)) represents Faraday's constant, M represents the molecular weight (kg/mol), and $i_c$ and $i_a$ represent the cathode and anode local current density (in $A/m^2$), respectively.

The water production model can be written as:

$$S_{H_2O} = \frac{M_{H_2O}}{2F} i_c \tag{14}$$

Due to the extra activation, heat generation is produced in the cell, and this causes changes in entropy and irreversibility:

$$\dot{q} = \left( \frac{-T \times \Delta S}{n_e \times F} + \eta_{act,c} \right) i_a \tag{15}$$

where $\dot{q}$ (in $W/m^2$) is the generated heat, $s$ (in J/(mol·K)) is the specific entropy, and $\eta_{act}$ and $n_e$ represent activation overpotential and the number of transferred electrons, respectively.

Employing the Butler–Volmer equation, the catalyst layers' local current density in both cathode and anode can be written as:

$$i_c = i_{o,c}{}^{ref} \left[ \frac{C_{O_2}}{C_{O_2}^{ref}} \right] \left[ \exp\left( \frac{a_a \times F}{RT} \times \eta_{act,c} \right) + \exp\left( -\frac{a_c \times F}{RT} \times \eta_{act,c} \right) \right] \tag{16}$$

$$i_a = i_{o,a}{}^{ref} \left[ \frac{C_{H_2}}{C_{H_2}^{ref}} \right] \left[ \exp\left( \frac{a_a \times F}{RT} \times \eta_{act,a} \right) + \exp\left( -\frac{a_c \times F}{RT} \times \eta_{act,a} \right) \right] \tag{17}$$

where, $C_{O_2}$ and $C_{H_2}$ are the concentrations of both oxygen and hydrogen respectively (in $mole/m^3$), $C_{O_2}^{ref}$ and $C_{H_2}^{ref}$ are the reference concentrations of both oxygen and hydrogen respectively (in $mole/m^3$), $i_{o,c}$ and $i_{o,a}$ are the reference densities of exchange current in both cathode and anode respectively, R = 8.314 represents the universal gas constant (in J/(mole·K)), and $a_c$ and $a_a$ are coefficients of charging transfer in both cathode and anode, respectively.

### 2.1.4. Membrane

The overall flow of water via the membrane is triggered by the equilibrium between the osmotic electrolytic withdrawal of water from the anode to the cathode and in reverse diffusion from the cathode to the anode:

$$N_w = n_d \frac{i}{F} - \frac{\rho_{mem}}{M_{mem}} D_w \nabla C_w - \frac{k_m}{\mu_l} \nabla P_{c \to a} \tag{18}$$

where $N_w$ (in kg/(m·s)) is the overall water flux throughout the membrane, $\rho_{mem}$ (in kg/m$^3$) is the dry membrane's density, $M_{mem}$ (in kg/mole) is the dry membrane's equivalent weight, $k_{mem}$ (in W/(m·K)) is the thermal conductivity of the membrane, $\mu_l$ (in kg/(m·s)) is the viscosity of liquid water, $C_W$ is the membrane's water content, $n_d$ is the electro-osmotic drag coefficient, $k_m$ (in m$^2$) is the membrane's hydraulic permeability, and $P_{c \to a}$ (in Pa) is the difference between both cathode and anode gas-phase pressures.

It can be taken into consideration that the membrane is a solid conductive material for transferring the heat as heat transfer undergoes through the membrane. By neglecting the energy transfer that relates to the overall water flow via the membrane, then:

$$\nabla \cdot [k_{mem} \nabla T] = 0 \tag{19}$$

Due to the proton's transfer resistance, the potential membrane loss can be represented as:

$$\nabla \cdot (\lambda_m \nabla \varphi) = 0 \tag{20}$$

where $\lambda_m$ (in S/m) is the membrane's ionic conductivity.

Nafion®117 was the type of membrane utilized for separating between the cathode and anode. The following mathematical concept for the ionic conductivity of the totally wetted membrane was introduced as [25]:

$$\lambda_m = \frac{Z_f \times D_H^+ \times C_f \times F^2}{R \times T} \tag{21}$$

where $Z_f$ is the fixed-site charge, $D_H^+$ (in m$^2$/s) is the protonic diffusion coefficient, and $C_f$ (in mole/m$^3$) is the fixed charge concentration.

### 2.1.5. Cell Potential

When the current is withdrawn, the consuming work of electrical energy is calculated from the FC. Therefore, the actual cell ($E_{cell}$) potentials are reduced from the potential dynamic equilibrium (E) due to the irreversible losses. Therefore, the cell potentials are obtained by subtracting all the excess quantities (losses) in the thermodynamic equilibrium potentials equation, as follows:

$$E_{cell} = E - [\eta_{act} + \eta_{mem} + \eta_{Ohm} + \eta_{diff}] \tag{22}$$

The Nernst equation can be used to determine the equilibrium potential, as follows:

$$E = 1229 \times 10^{-3} - \left[83(T - 298.15) + 4.3085T \left[\ln\left(P_{H_2} \times P_{H_2}{}^{0.5}\right)\right]\right] \times 10^{-5} \tag{23}$$

Butler–Volmer Equations (16) and (17) determine the overpotentials of anode and cathode activation, where Equations (11) and (20) represent the calculation of the overpotentials in GDLs and proton overloads in the membrane, respectively.

The following equations provide the calculation of the overpotentials of the anode and cathode spread [26]:

$$\eta_{diff,\,c} = \frac{RT}{F} \ln \sqrt{\left(1 - \frac{i_c}{i_{L,c}}\right)} \tag{24}$$

$$\eta_{\text{diff, a}} = \frac{RT}{F} \ln \sqrt{\left(1 - \frac{i_a}{i_{L,a}}\right)} \tag{25}$$

$$i_{L,c} = \frac{2F \times C_{O_2} \times D_{O_2}}{\delta_{GDL}} \tag{26}$$

$$i_{L,a} = \frac{2F \times C_{H_2} \times D_{H_2}}{\delta_{GDL}} \tag{27}$$

### 2.2. Boundary Conditions

The feature of the cell rotating geometry is used to reduce the computational cost, and symmetry is assumed in the y-direction, so all gradations in the y-direction are set to be zero at the limits of the plane x-z except for inlets and outlets of the channel. Moreover, the condition of zero flux in the x-boundaries was applied for y-z planes. A description of the velocity, temperature, and species concentrations has been provided, which are the anode and cathode inlet DBCs values. Based on the current density, the air and fuel inlet speeds can be calculated as:

$$V_{in,c} = \alpha_c \left(\frac{I}{4F}\right) \times A_{MEA} \times \frac{RT_{in,c}}{X_{O_2,in} \times P_{c, in} \times A_{ch}} \tag{28}$$

$$V_{in,a} = \alpha_a \left(\frac{I}{2F}\right) \times A_{MEA} \times \frac{RT_{in,a}}{X_{H_2,in} \times P_{a, in} \times A_{ch}} \tag{29}$$

where $\alpha$ is the stoichiometric flow ratio, AMEA is the area of the MEA (in m$^2$), and $A_{ch}$ is the flow channel cross-sectional area (in m$^2$).

The outlet pressure for the gas flow channels is the needed pressure for the electrode. The flow variation in the x-direction is considered to be negligible according to NBCs.

The temperature value is specified for the upper as well as lower surface areas of the cell, which are the outside surface areas in the z-direction, and also there is no heat addition related to the x-y plane of the performing boundary surface areas. Both DBCs and NBCs can be effectively utilized to examine the protonic as well as electronic potential formulas for the system. DBCs are applied at the smoothed ground surface area, while the NBCs are executed at the interface between the layers of gas diffusion as well as gas channels to establish that there is no potential flux condition into the gas channels. Likewise, the protonic potential field needs absolutely no flux BCs and also a group of potential BCs provided at the CCL and ACL interfaces, respectively [27,28].

### 2.3. Procedure of Solution Algorithm

The CFD Multi-Physics code was utilized to separate governing equations, making use of FEM. In order to make certain that the solutions were independent of the dimension of the network, comprehensive computational analyses were carried out. A quadratic mesh with a total of 64,586 nodes and 350,143 meshes was utilized to obtain adequate spatial resolution [29]. In order to measure the entry flow rate on both sides of the cathode and anode, the proper current density needs to be computed. An initial approximation of the potential for activation of the excess current density is acquired employing the Butler–Volmer equation. Additionally, the mass fractions of hydrogen, oxygen, nitrogen, and water were obtained by calculating the flow fields for the velocities in the 3 directions (x, y, and z), along with the pressure, P. Lastly, in the series of transportation equations for the cell's temperature and potential fields in the membrane and the layers of gas diffusion, scalar equations have actually been resolved. The current regional density and the regional excess activation potential were additionally identified relying on the Butler–Volmer equation. Regional overpotential activation is modified for each global iterative loop. On each variable, the convergence conditions are executed, and the procedure proceeds until convergence is achieved. The model conducts parametric research studies as well as checks out the influence of different specifications on the efficiency of transportation

strategies and fuel cells, and this is one of the most essential strengths of the considered model. The main algorithm feature built in this study was the capability of the algorithm for exact evaluation of the regional activation overpotentials, where it consequently causes boosted prediction of the distribution of regional current densities. Figure 1 shows the algorithm's flow diagram. Table 1 lists the specifications for the basic case simulations.

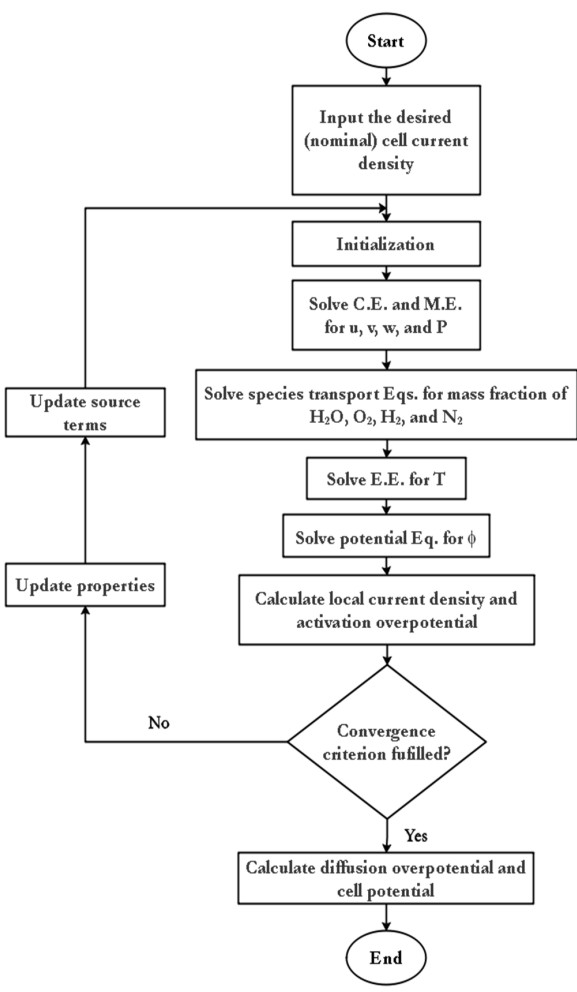

**Figure 1.** Flow diagram of the solution's algorithm.

**Table 1.** Geometrical, operational, electrode, and membrane parameters.

| Parameter | Symbol | Value | Reference |
|---|---|---|---|
| Channel length (m) | L | 0.05 | |
| Channel width (m) | W | $1 \times 10^{-3}$ | |
| Channel height (m) | H | $1 \times 10^{-3}$ | |
| Land area width (m) | $W_{land}$ | $1 \times 10^{-3}$ | |
| Gas diffusion layer thickness (m) | $\delta_{GDL}$ | $26 \times 10^{-5}$ | |
| Membrane thickness (Nafion® 117) (m) | $\delta_{mem}$ | $2 \times 10^{-4}$ | |
| Catalyst layer thickness (m) | $\delta_{CL}$ | $2 \times 10^{-5}$ | |
| Hydrogen reference mole fraction | $X_{H_2}^{ref}$ | 0.84639 | |
| Oxygen reference mole fraction | $X_{O_2}^{ref}$ | 0.17774 | |
| Anode pressure (atm) | $P_a$ | 3 | |
| Cathode pressure (atm) | $P_c$ | 3 | |
| Inlet fuel and air temperature (K) | $T_{cell}$ | 353.15 | |
| Relative humidity of inlet fuel and air (%) | $\varphi$ | 100 | |

**Table 1.** *Cont.*

| Parameter | Symbol | Value | Reference |
|---|---|---|---|
| Air stoichiometric flow ratio | $\alpha_c$ | 2 | |
| Fuel stoichiometric flow ratio | $\alpha_a$ | 2 | |
| Heat transfer coefficient between solid and gas phase (estimated) (W/m$^3$) | $\beta$ | $4 \times 10^6$ | [24] |
| Protonic diffusion coefficient (m$^2$/s) | $D_{H^+}$ | $4.5 \times 10^9$ | [25] |
| Fixed-charge concentration (mol/m$^3$) | $C_f$ | 1200 | [25] |
| Fixed-site charge | $Z_f$ | $-1$ | [25] |
| Electrode porosity | $\varepsilon$ | 0.4 | [25] |
| Membrane ionic conductivity (humidified Nafion® 117) (S/m) | $\lambda_m$ | 17.1223 | [25] |
| Electrode hydraulic permeability (m$^2$) | $k_p$ | $1.76 \times 10^{-11}$ | [28] |
| Electrode electronic conductivity (S/m) | $\lambda_e$ | 100 | [30] |
| Transfer coefficient, anode side | $a_a$ | 0.5 | [31] |
| Transfer coefficient, cathode side | $a_c$ | 1 | [32] |
| Cathode reference exchange current density (A/m$^2$) | $i_{o,c}^{ref}$ | $1.8081 \times 10^{-3}$ | [33,34] |
| Anode reference exchange current density (A/m$^2$) | $i_{o,a}^{ref}$ | 2465.598 | [33,34] |
| Electrode thermal conductivity (Ballard AvCarb® -P150) (W/m·K) | $k_{eff}$ | 1.3 | [35] |
| Membrane thermal conductivity (W/m·K) | $k_{mem}$ | 0.455 | [35] |
| Entropy change of cathode side reaction (J/mol·K) | $\Delta S$ | $-326.36$ | [36] |
| Electro-osmotic drag coefficient | $n_d$ | 2.5 | [37] |

### 2.4. PEM Fuel Cell Stack and Plant Design

The fuel cell stack is made to consist of two current plates, two bi-polar plates, two gaskets, two GDLs, two endplates, and an MEA (membrane layer with addition to two CLs). Both the anode and the cathode have direct gas flow channels in a cross-sectional area of 1 mm$^2$. A serpentine water flow channel lies on the top side of the bipolar cathode plate. The material characteristics and dimensions of for each element are displayed in Table 2.

**Table 2.** Material type and dimensions of the stack components.

| Property | MEA | GDL | Bipolar Plate | Current Collector | Gasket | Endplate |
|---|---|---|---|---|---|---|
| Material | Nafion® | Carbon paper | Carbon graphite | C15720 copper | Silicon® | Stainless steel |
| Density (kg/m$^3$) | 2000 | 400 | 1800 | 8700 | 2330 | 7800 |
| Dimensions (mm) | $330 \times 330$ | $300 \times 300$ | $350 \times 350$ | $350 \times 350$ | $330 \times 330$ | $400 \times 400$ |
| Thickness (mm) | 0.24 | 0.26 | 4 | 2 | 0.26 | 20 |

The designed voltage and current at the maximum power of the PEM fuel cell operation are selected from Figures 2 and 3, as 0.61 volt and 1.04 Ampere/cm$^2$, respectively.

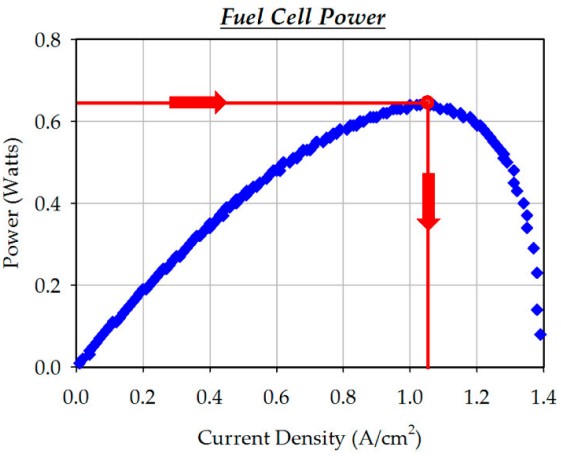

**Figure 2.** PEM fuel cell polarization curve (selected voltage corresponded to the current at peak power).

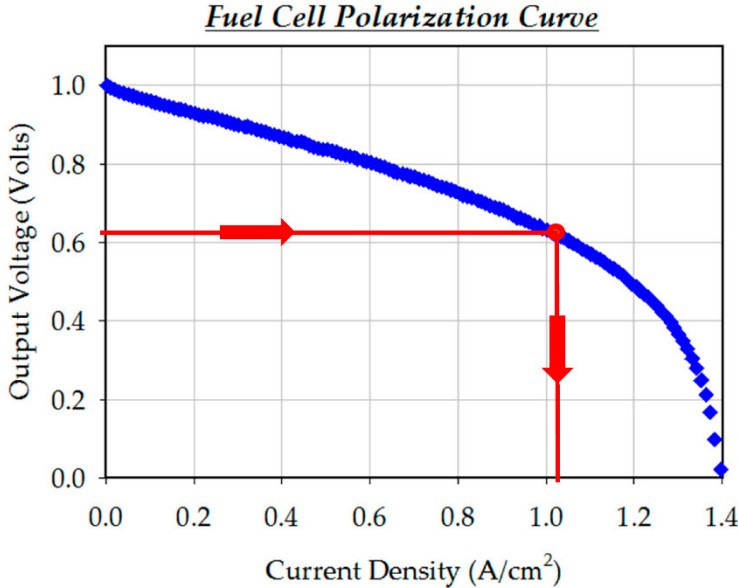

**Figure 3.** PEM fuel cell polarization curve (selected voltage corresponded to the current at peak power).

The power density produced from this operating point can be written as:

$$\text{Power (in W)} = I \times V \tag{30}$$

Area of one channel = 6 cm$^2$
Number of channels for one cell layer = 150
Area for one-layer cell = 900 cm$^2$
The total power of one layer = 570.96 Watt
No. of layers in one stack is designed to be = 13 layers
The total power of one stack = 7422.48 Watt
The mass of one stack = 62.08 kg
The power required from fuel cell power plant = 1500 kW
No. of stacks required for the power plant = 203
The following equation can be used for the hydrogen consumption calculation:

$$\dot{m}_{H_2} = \frac{M_{H_2} \times I}{2F} \times 3600 \tag{31}$$

where, $M_{H_2}$ is molecular mass of hydrogen equal to $2.02 \times 10^{-3}$ (kg/mole), and F = 96,485 Coulombs/electron-mole (Faraday's constant).

## 3. System Description

In our studied hybrid system, we need some basic sensitivity variables for design to improve component sizes and system cost, where the application must be evaluated before designing and simulating the system and knowing some parameters, such as location, load, and solar radiation.

### 3.1. Case Study

The researched hybrid power system was performed in the Bahr AL-Najaf area, where the examination was performed based upon a unique profile for the region of actual load and also various climate condition specifications. The latitude and longitude for the selected location were 32 °N latitude, 44 °E longitude. The estimated lifetime is approximately 25 years, while the yearly rates of interest are taken as about 8%.

The Bahr AL-Najaf region load was already set at 12,000 kWh/day. The maximum load on a seasonal scale is assumed to be 1.5 MW. It must be observed that the yearly maximum load of 1.5 MW happens in June. The peak season (between June and July) has the highest demand, while the low season (between December and January) has the lowest demand.

### 3.2. Electrical Load Profile

Figure 4 reveals the change in hourly average loads during the year in the Bahr Al-Najaf region. The yearly highest load in the seasonal range is 1.5 MW. This load takes place in June, complied with by July.

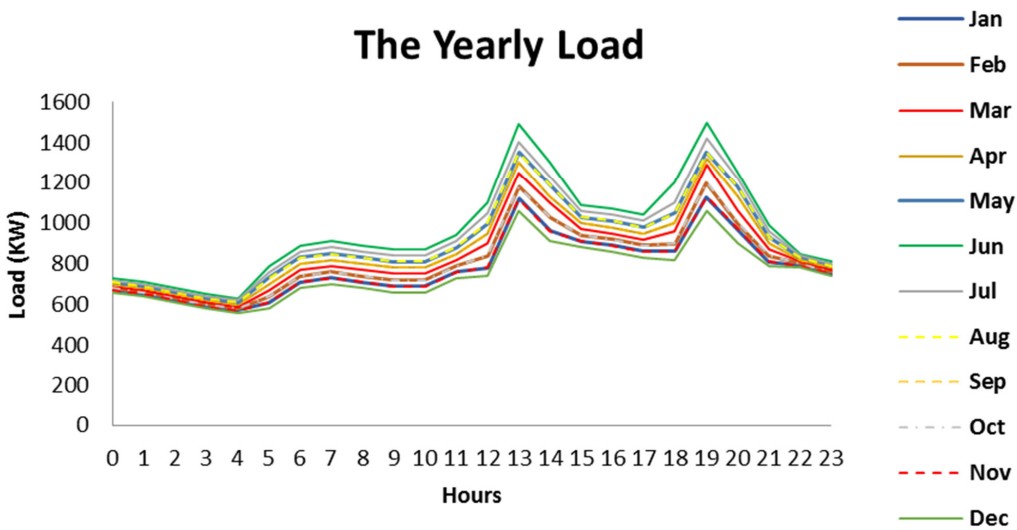

**Figure 4.** Variation of hourly average load for every month during the year.

### 3.3. PV Generator and Solar Resource Information

For the Najaf Sea region, solar radiation information is recorded in this paper from NASA's Atmospheric Data Center [37,38]. Figure 5 reveals the regular monthly average of solar power information along with global horizontal irradiation data during 2020. The average yearly solar radiation for the area studied is 5.16 kWh/m$^2$/day. The PV generator was oriented with azimuth angle equal to 0° with the south direction, and a 32° angle with the horizontal direction. According to the criteria set by the system, the lifetime of the PV system is 25 years, with a de-rating factor of 80% and also a ground reflectance of 20%. Table 3 reveals the average quality and solar irradiance index considering the impact of the temperature level of the PV matrix.

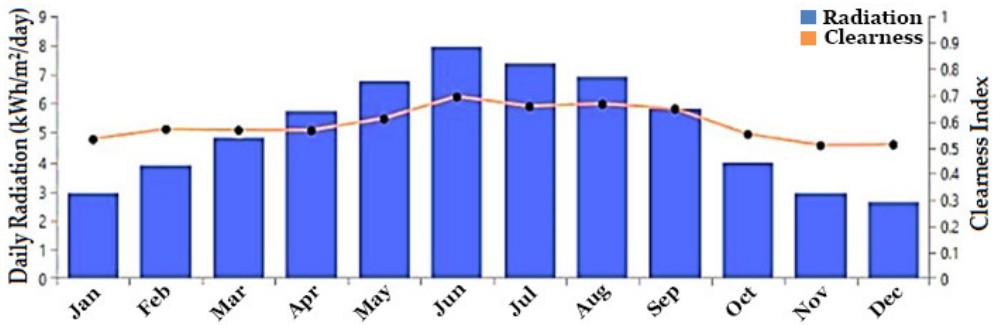

**Figure 5.** Monthly average solar global horizontal irradiance (GHI) data during 2020.

**Table 3.** Index of clarity and mean solar radiation.

| Month | Clearness Index | Average Radiation (kWh/m$^2$/day) |
|---|---|---|
| January | 0.530 | 2.960 |
| February | 0.568 | 3.910 |
| March | 0.565 | 4.860 |
| April | 0.564 | 5.730 |
| May | 0.608 | 6.770 |
| June | 0.692 | 7.940 |
| July | 0.655 | 7.370 |
| August | 0.664 | 6.920 |
| September | 0.646 | 5.820 |
| October | 0.549 | 4.000 |
| November | 0.507 | 2.950 |
| December | 0.510 | 2.640 |

### 3.4. Wind Turbine Generator

Wind turbine characteristics in our system were researched. The operating life for the wind turbine is 20 years, while the generated energy is 1500 kW. The WT's hub height was 80 m and there is no overall loss. Figure 6 reveals the wind output power of the used wind turbine via the variation of wind speed.

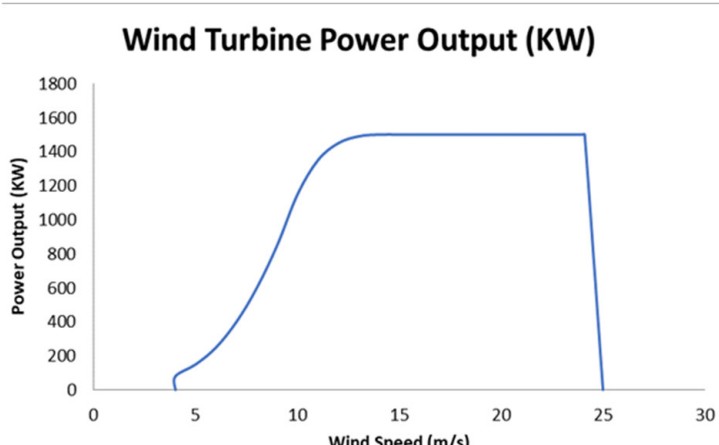

**Figure 6.** The wind turbine's output power.

### 3.5. DC/AC Inverters Design

Fuel cells and PV produce DC electric voltage. Due to the industrial and household loads being mostly AC loads, therefore, a power converter is required to transform the DC voltage produced by the fuel cell into AC voltage. The inverter consists of a power stage with IGBT or MOSFET transistors, which chop the DC voltage so that an AC voltage signal is obtained. Inverters can be single-phase or three-phase voltage depending on the characteristics and power required by the load demand. Figure 7 shows the configuration of the used DC/AC inverters. For FC:

$$DC_{power} = AC_{power} \times (1 - LF) \tag{32}$$

$$I_{DC} \times V_{DC} = I_{AC} \times V_{AC} \times (1 - LF) \tag{33}$$

where, V is voltage, I is current (A), and LF is the losses factor and it is equal to 0.1.

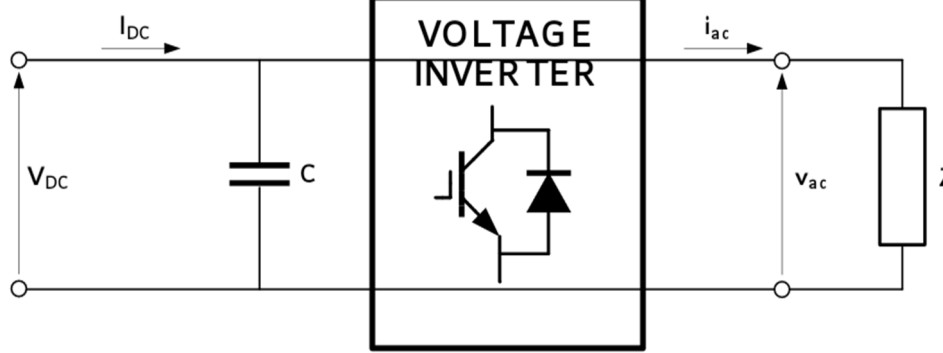

**Figure 7.** DC/AC inverter design.

For PV:

$$I_{DC} \times V_{DC} = \frac{1}{\sqrt{3}} \times P \times PF \times V_{AC} \qquad (34)$$

where, PF is power factor, and it is equal to 0.1–0.9.

### 3.6. Hybrid System Modeling

The studied hybrid system consists of a PV matrix, a hydrogen-fueled system, and a wind turbine. In hybrid energy systems, we use fuel cells with an off-grid hydrogen tank. In this paper, we performed an improvement process by defining the decision variable by HOMER Pro, taking into account all values. Figure 8 represents the HOMER Pro model of the studied hybrid energy system. The process of improving values includes selecting the variable of the decision that the designer chooses. Accordingly, the control is optimized, as HOMER Pro is based on probability values in the process of improvement. The system includes the following variables:

- Array size of PV;
- Size of FC;
- Size of wind turbine;
- Size of DC/AC converter;
- Size of hydrogen tank and electrolyzer.

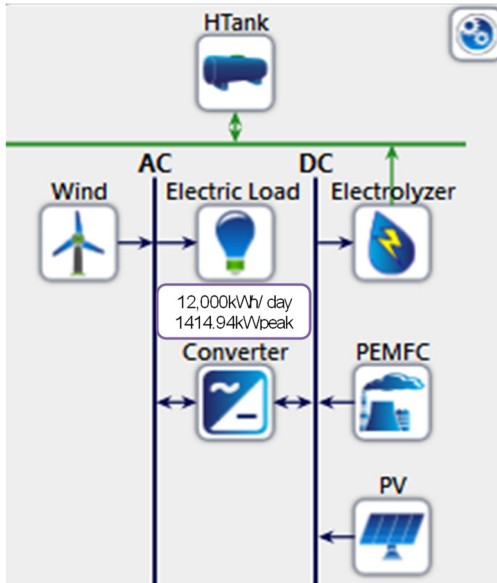

**Figure 8.** PV/FC/Wind hybrid power system.

The studied hybrid system must be sufficient for the electrical load at a rate of 100%. The yearly actual interest rate is about 0.2%, which represents the nominal interest rate minus the rate of inflation.

Additionally, the model includes the limit of the yearly maximum capacity shortage equal to 0%, and the operating reserve equal to 10% of the hourly load.

### 3.7. Operational Control Strategy

The operational control strategy is illustrated in normal operating conditions. The load demand is covered by PV. When a surplus in demand occurs, it will be used to feed the electrolyte for obtaining and storing the hydrogen. In that case, the tank is completely full, then the energy is transferred to a discharge load. If that energy is less than the consumption, the fuel cells will produce the energy to cover the demand, and the fuel cells must cover the demand completely in the absence of radiation. A wind turbine has been integrated into the proposed system to make it more reliable. The studied system gives a clear vision of how PV, wind turbines, and fuel cells all compete together in an integrated, independent application. Table 4 shows the selected HOMER Pro input data [39].

**Table 4.** Input data on option costs.

| Options | Capital Cost | Replacement Cost | O&M Cost | Lifetime |
|---------|-------------|------------------|----------|----------|
| PV | 20,000 USD/100 kW | 20,000 USD | 5 USD/year | 25 years |
| Wind | 875,000 USD/1500 kW | 875,000 USD | 3000 USD/year | 20 years |
| Fuel Cell | 2400 USD/7.5 kW | 2000 USD | 0.08 USD/h/kW | 40,000 h |
| Converter | 2000 USD/50 kW | 200 USD | 100 USD O&M/year | 20 years |
| Electrolyze | 1000 USD/10 kW | 100 USD | 10 USD/year | 15 years |
| Hydrogen | 1000 USD/2000 kg | 100 USD | 10 USD/year | 20 years |

## 4. Results and Discussion

### 4.1. PEM Fuel Cell Analysis

As noted in Figures 2 and 3, the nominal current density at maximum power was equal to 1.04 A/cm$^2$. Therefore, the following analysis results are for cells operated at this nominal current. Figure 9 demonstrates the variation of the velocity profiles along the mid-plane of the CGFC and AGFC at maximum power. It was observed that the velocity profile of the maximum velocity remains in the center of the channels and uses the same specific pattern until reaching the end of the channels. It is clear that the entry velocity into the CGFC is much greater than that of the AGFC. It is shown from Figure 9 that the maximum velocity in the mid-plane of CGFC was 0.0604 m/s, while the maximum velocity in the mid-plane of AGFC was about 0.03 m/s. The reason is that the molar fraction of hydrogen in the wet inlet gas is higher than that of oxygen (Equations (28) and (29)).

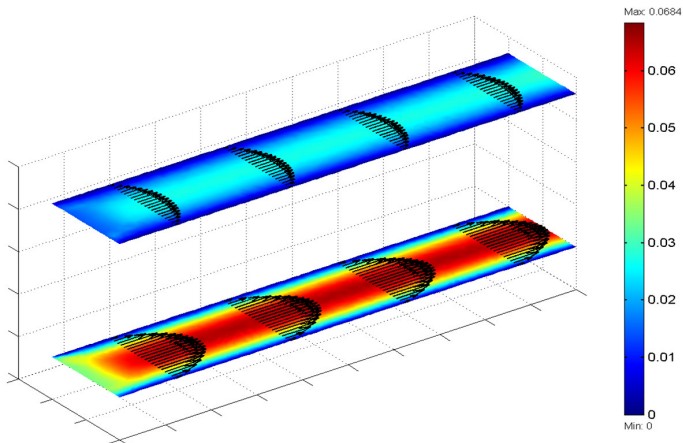

**Figure 9.** Velocity profiles in the mid-plane of the cathode and anode gas flow channels at peak power.

Figure 10 displays the 2D (y-z plane) velocity vectors in the anode and cathode GDLs at x = 25 mm. The reactions on the cathode catalyst layer (CCL) lead to absorb oxygen that creeps up on the surface, causing production of water vapor.

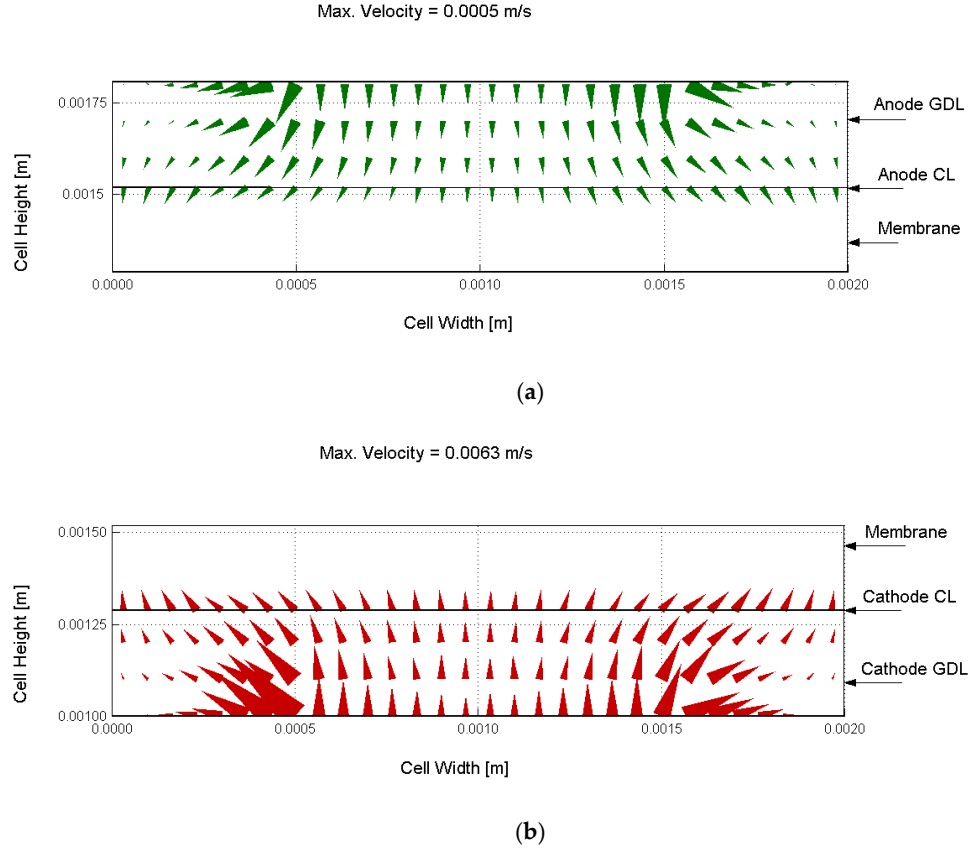

**Figure 10.** Velocity profiles of: (**a**) the anode GDL and cathode GDL, and (**b**) the y-z plane at x = 25 mm.

On the other side, the vectors are guided down the GDL anode (directed to the surface of the membrane) and dispersed across the diffusion layer, and then absorbed on the surface of the anode catalyst layer (ACL). This is due to the reactions of H2 and water transfer through the membrane. It was clearly shown that the velocity in the AGDL is much less than that in CGDL, which is due to the molar fraction of oxygen in the wet inlet gas being less than that of hydrogen.

The in-depth distribution of oxygen molar fraction at maximum power is displayed in Figure 11. Due to the oxygen usage in the catalyst layer, the oxygen concentration gradually decreases from the flow channel to another channel. Therefore, it was seen that the molar fraction of oxygen at the entrance of FC was about 0.178, and it decreases until it reaches the value of 0.027 at the exit of the channel. The amount of oxygen under the land area in the GDL is lower than that under the channel area. Oxygen is decreased and its percentage is diffused in the direction of the layer of the catalyst, so its concentration is stabilized by a steady concentration in the layer of the catalyst. Considerable depletion of oxygen under land areas is triggered by the reduced diffusion of oxygen with a reduced concentration in the surrounding air. The consumption rate of oxygen is reduced sufficiently so as to not create diffusive constraints at a reduced current density, and also the oxygen concentration has actually already become near zero. Furthermore, the concentration of oxygen is the factor on which the regional current density depends directly on the reaction side of the cathode. This indicates that, especially near the outlet, the regional current density distribution under the land areas is much lower than under the channel regions. The major barrier to the current reaching its high density is the diffusion of oxygen in the direction of the catalyst layer.

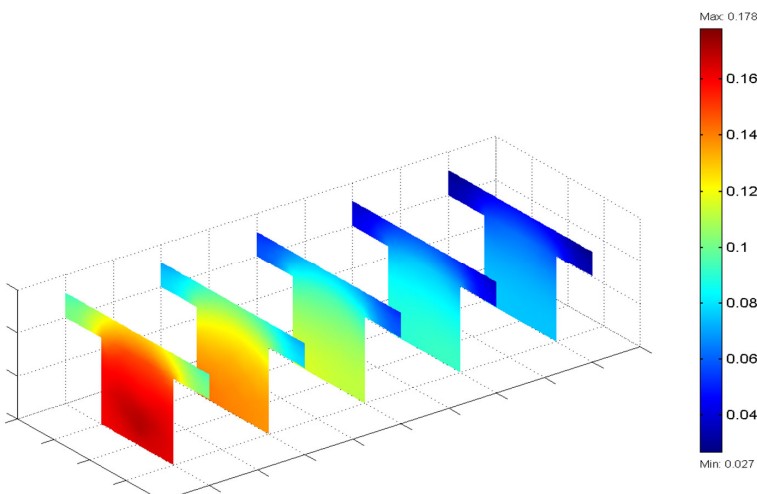

**Figure 11.** Variation of oxygen molar fraction within the cathode at peak power.

As a result of the reasonably reduced diffusion of oxygen contrasted to hydrogen, the raised oxygen consumption causes the current increase in density. The operating conditions for the cathode identify the current density of the FC when working in humidified air. The variation of the hydrogen molar fraction on the anode side at maximum power is displayed in Figure 12. It was shown that the hydrogen molar fraction varied from its maximum value of 0.846 at the inlet of FC on the side of the anode to the minimum value of 0.753 at the exit of FC. Therefore, throughout hydrogen consumption, its concentration reduces from the inlet to the outlet, which causes an extremely small reduction along the channel because of the high diffusion of hydrogen. The decline in hydrogen moles concentration below land areas is less than that for oxygen on the side of the cathode.

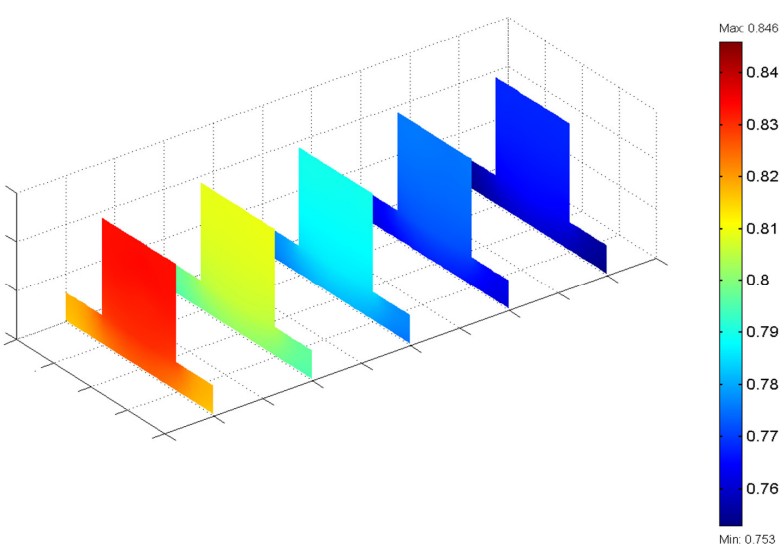

**Figure 12.** Variation of hydrogen molar fraction within the anode at peak power.

The distribution of the water molar fraction in the cell at the maximum power is shown in Figure 13. Noticeable condensation was estimated to take place within the cathode region for the small current densities. It was seen that the highest mole fraction of the water vapor, in this case, overtakes the saturated value. To make sure, it is indicated that the condensation of the vapor took place. Nevertheless, the mole fraction of water for the high current density is greater than that for the low current density, while the back diffusion is dominated by electro-osmotic effect under high nominal current density. These phenomena occur with a greater current density in a dryer anode.

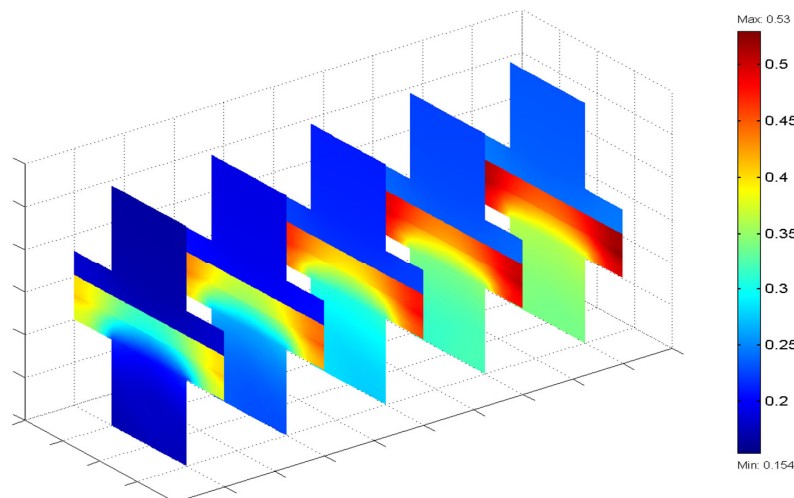

**Figure 13.** Variation of water molar fraction within the cell at peak power.

It is necessary to remove the heat from electrochemical reactions since approximately 50% of this energy is released during the maximum energy density. The reason to remove this energy is to avoid damage to the membrane as a result of high thermal stresses in addition to drying the membrane. In PEM fuel cells, thermal regulation is a difficult issue owing to the slight temperature difference between the working environment and the fuel cell stack. Knowing the temperature distribution during the study of the fuel cell has beneficial effects and important implications for the understanding of all transport phenomena. Thus, it leads to knowledge of the values of increases in temperature rates and the possibility of avoiding failure of the parts of the cell

Figure 14 indicates the temperature distribution at maximum power conditions. It is clear that the temperature at the anode is lower than that in the anode. The reason for this is the irreversible and reversible processes in entropy production. The maximum temperature is achieved near the entrance to the cathode as the electrochemical activity is at its highest. The main heat generation takes place in the CCL because it is the region where the temperature peak appears. As a result of thermal energy generation in the CCL, a high current is generated in this region. It was observed that the temperature distributions are identical for all loading conditions. The maximum temperature exceeded the inlet gas temperature, reaching higher than about 7000 K in the CCL region.

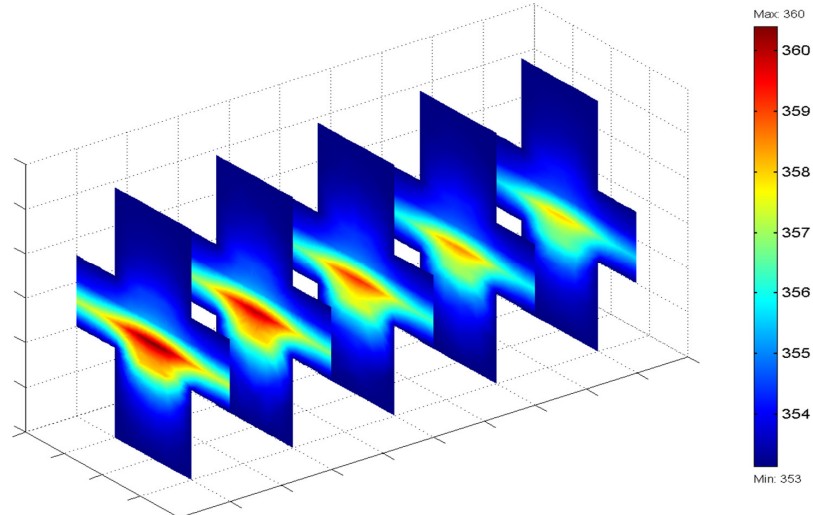

**Figure 14.** Temperature profile within the cell at peak power.

Figure 15 illustrates the variation of regional current density at the CCL at peak power. The regional current densities have actually been stabilized by the small current density (i.e., $i_c/I$). The maximum current fraction was about 140%. It can be observed that the use of the catalyst under the land areas is due to the fact that a high fraction of the current is created at the catalyst layer that exists below the channels for a high nominal current density. A consistent current density generation is preferable for optimum FC performance. This can be attained with a non-uniform catalyst distribution, perhaps combined with non-homogeneous GDES [34,35].

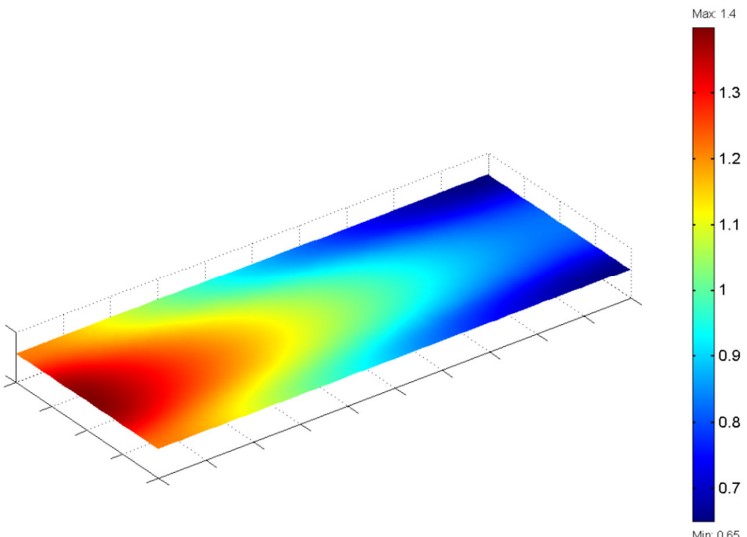

**Figure 15.** Variation of dimensionless local current density ($i_c/I$) at CCL at peak power.

The variance of the cathode activation overpotentials (in volts) is presented in Figure 16. There are more significant values below the channel region, where the distribution patterns of activation overpotentials are identical. It can be easily found that the activation overpotential profile correlated with the regional current density. The reason for this is that the current densities accompany the highest possible reactant concentrations, where it is at the highest value in the middle of the channel.

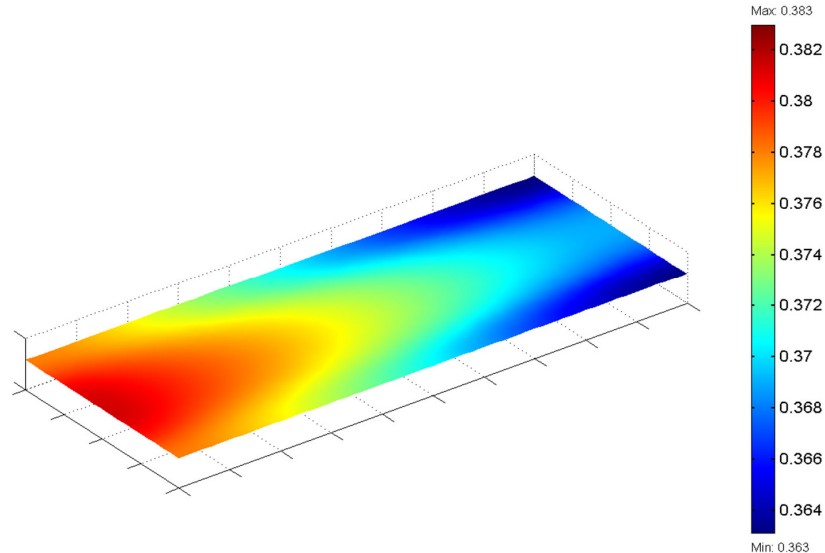

**Figure 16.** Variation of activation overpotential at the CCL at peak power.

Ohmic overpotential is the loss related to resistance to electron transportation in the gas diffusion layers. The size of this overpotential depends on the pathway of the electrons, for a provided small current density. The potential field (in volts) in the anodic and the cathodic gas diffusion electrodes are displayed in Figure 17. The potential variation is regular to the flow channel and the sidewall surfaces, while there is a decrease in the land region where electrons move into the bipolar plate. Moreover, it was observed that the ohmic losses are much higher in the location of the catalyst layer under the flow channels, where the variation shows a decrease in both x- and y-directions as a result of the non-uniform regional current creation.

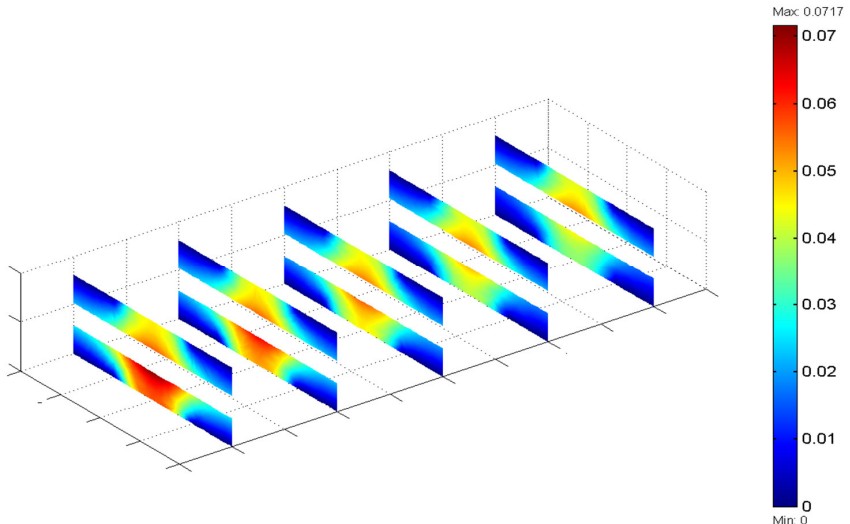

**Figure 17.** Variation of Ohmic overpotential in the AGDL and CGDL at peak power.

The potential losses inside the membrane are caused by a resistance to the protons' transportation throughout the membrane from the ACLs to the CCLs. The pathway that protons travel and activities in the catalyst layers are the source of protonic overpotential variation. Figure 18 indicates the potential loss variation in the membrane (in Volts). It was observed that the decreased potential is consistently distributed. There is a low gradient in the distribution of concentration of hydrogen below the channel and over a large area at the ACL. From Figure 18, it was shown that the value of the membrane overpotential was varied in the range of 0.0005–0.219 Volts at the entrance of the FC, while it is varied in the range of 0.00050.1 Volts at the exit of the FC.

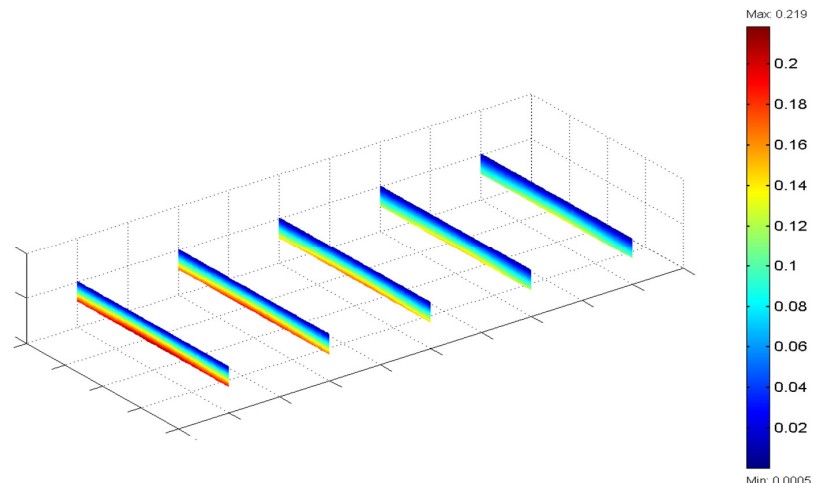

**Figure 18.** Variation of membrane overpotential throughout the membrane at peak power.

Figure 19 demonstrates the variation of the CCL diffusion overpotentials (in Volts). The diffusion overpotential variation patterns are identical, with a higher value in the channel region location. Additionally, the local current density correlates with the profile of overpotential diffusion. From Figure 19, it was observed that the overpotential in the channel region was varied from the minimum value of 0.008 Volts at the exit of the channel to the maximum achieved value of 0.0244 Volts at the entrance to the channel.

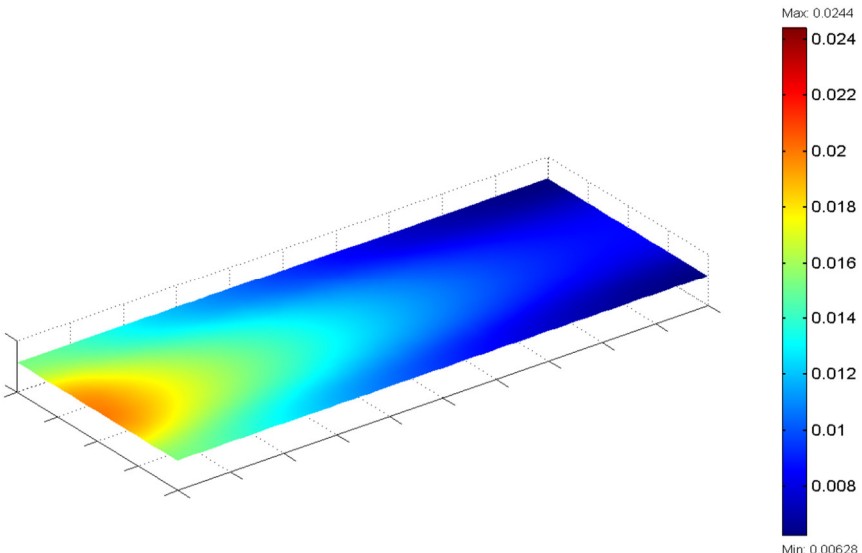

**Figure 19.** Variation of diffusion overpotential at the CCL at peak power.

*4.2. Results of Optimization*

Table 5 lists optimization findings from the HOMER software for the PV-FC/wind turbine hybrid system employing TNPC. The suggested system would provide the region of Bahr Al-Najaf with electrical power and is capable of covering the need for the continuity of the load throughout the year.

**Table 5.** The hybrid PV/FC/wind model's optimization results.

| | |
|---|---|
| PV (kW) | 18,000 |
| PEMFC (kW) | 1500 |
| Wind (kW) | 4500 |
| Converter (kW) | 1194 |
| Electrolyzer (kW) | 6400 |
| Hydrogen tank (kg) | 716,250 |
| Initial Capital Cost (USD) | 7,750,000 |
| Total NPC (USD) | 9,573,208 |
| COE (USD/kWh) | 0.169 |
| Operating cost (USD/year) | 140,941 |

The optimum design was achieved by running several calculations with solar radiation intensity of 5160 W/m$^2$/day and an overall clear annual index of 0.564, considering various FC, PV, wind turbines, hydrogen tank, transformer, and the electrolyzer.

A loading control strategy was used in this paper, and a TNPC Hybrid System was obtained, with 9,573,208 USD. In addition, the yearly capital and energy costs (COE) were 7,750,000 USD and 0.169 USD/kWh, respectively. Figure 20 summarizes the proposed hybrid energy system and the financial costs. Figure 21 indicates the average monthly output of energy for each renewable source. Table 6 displays the hybrid energy system's yearly emissions.

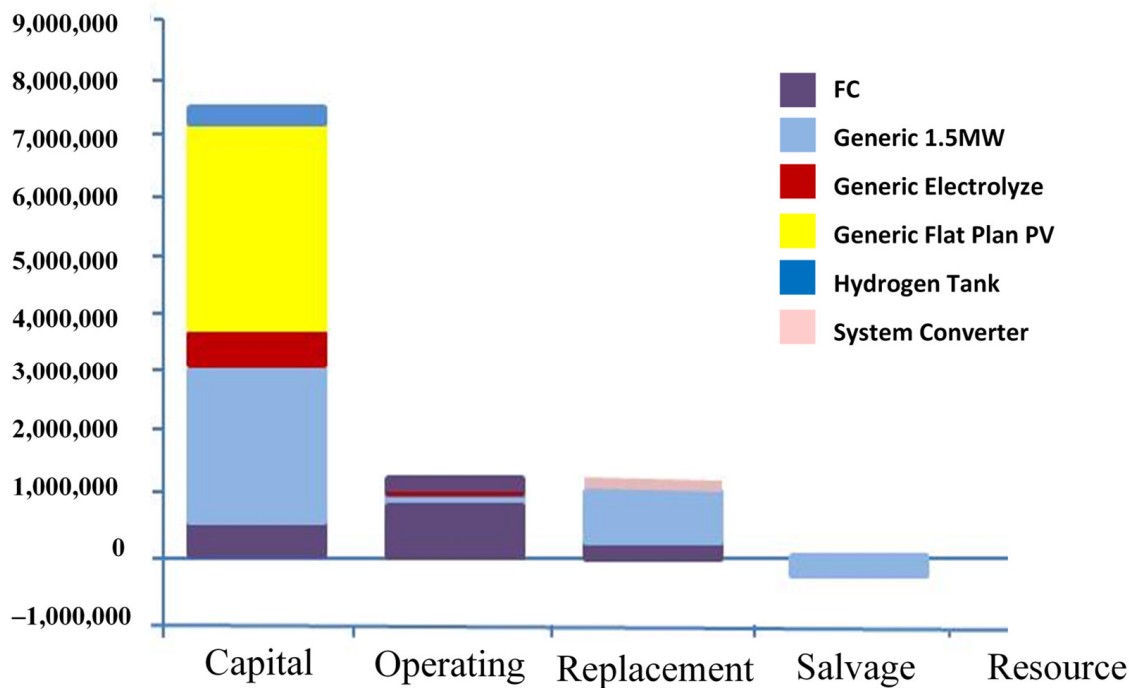

**Figure 20.** Summary of total costs of the PV/FC/Wind hybrid suggested system.

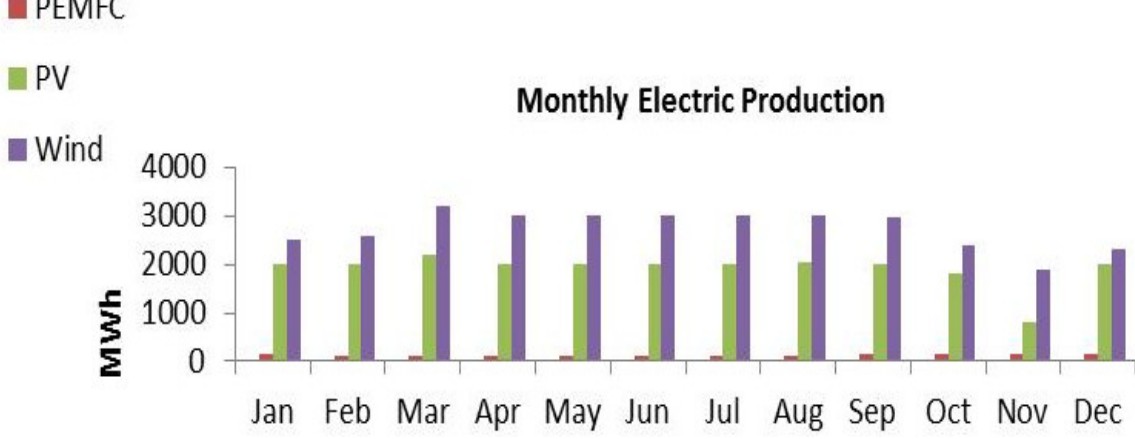

**Figure 21.** Monthly average electric production.

**Table 6.** The yearly emissions of the suggested system.

| Pollutant | Emissions (kg/year) |
| --- | --- |
| Carbon dioxide | −9748 |
| Carbon monoxide | 6203 |
| Unburned hydrocarbons | 329 |
| Particulate matter | 53.0 |
| Sulfur dioxide | 0 |
| Nitrogen oxides | 1189 |

Figures 22–24 reveal the yearly operating output of the PV, the FC generator, and wind turbine generator, respectively. These figures are provided to demonstrate the operational control strategy based on the region of Bahr AL-Najaf load demand and climate data.

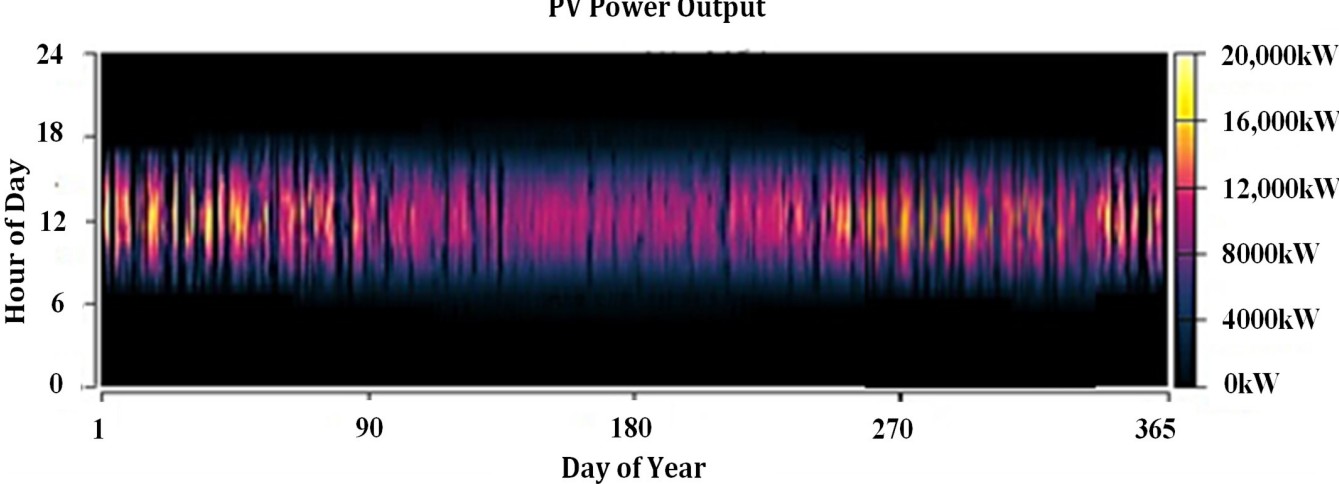

**Figure 22.** Annual PV generator operation.

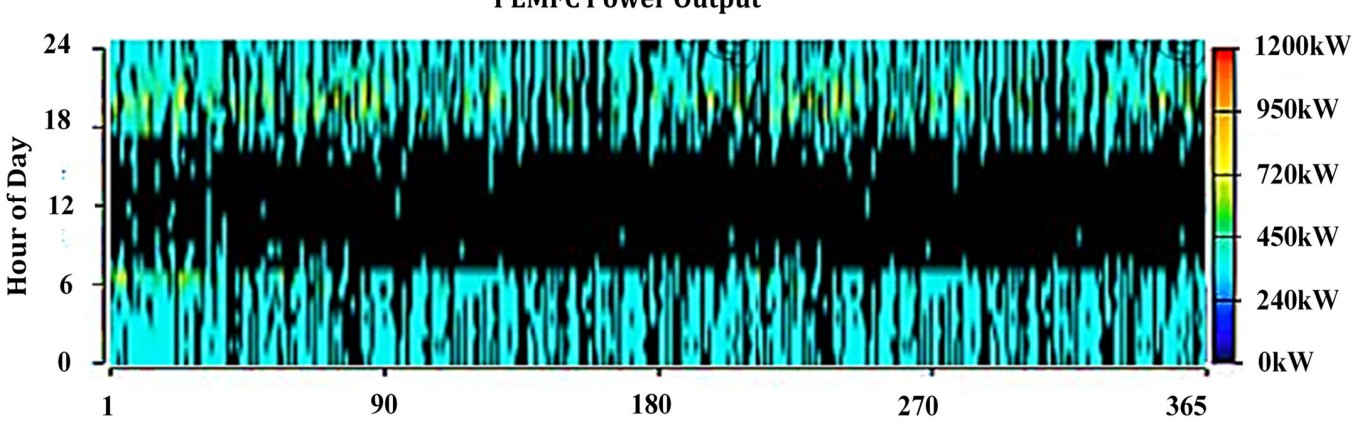

**Figure 23.** Annual PEMFC generator operation.

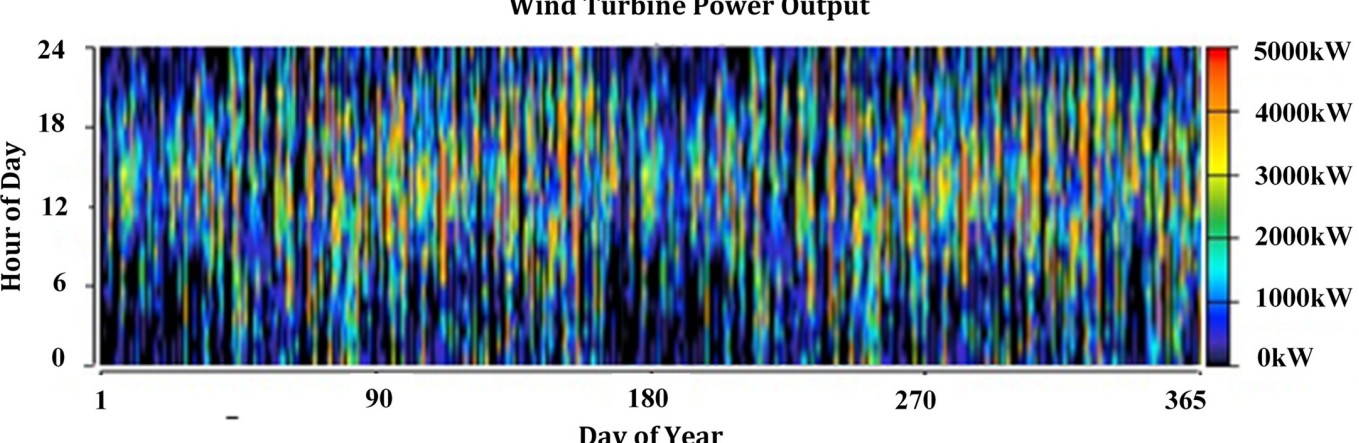

**Figure 24.** Annual Wind turbine generator operation.

## 5. Conclusions

This paper carried out a highly reliable hybrid system without battery storage to meet electric load demand in the Bahr AL-Najaf region, south of Iraq. This system consists of the following main components: wind turbine, fuel cell, PV, electrolyzer, inverter, and

storage tanks. The suggested model was focused on CFD simulation and the economic performance and was mostly dependent on the power supply loss prospect conception. The total net present cost analysis explicitly demonstrated that the suggested hybrid power system, specifically fuel cells, was a suitable replacement for diesel generators because it is a reliable source for the production of electrical energy in an environmentally friendly and pollution-free manner, with relatively low operating costs. It was also noticed that a fuel cell generator can effectively supplement a varying renewable energy source such as solar energy to meet rising loads. This study provided the analysis of the proposed hybrid power system to cover the electricity demand in the Bahr AL-Najaf region of 1.5 MW, 4.38% fuel cells, 26.3% wind turbines, and 69.3% PV. The cost of production per kilowatt is 0.169USD of energy. Moreover, it was found that the optimum design of the suggested HES was achieved by running several calculations with solar radiation intensity of 5160 W/m$^2$/day and an overall clear annual index of 0.564. In addition, it was concluded that the TNPC Hybrid System was obtained with 9,573,208USD, and the yearly capital and energy cost (COE) was 7,750,000 USD.

The following research topics can be considered worthy of further investigation for the study in this paper, modeling:

1. The effect of the channel geometrical configurations on the PEMFC mechanical behavior using CFD modeling.
2. The effect of seal (gasket) properties on the contact pressure distribution inside PEMFC using CFD modeling.
3. The effect of GDL porosity and the catalyst layer thickness on the performance of a PEMFC using CFD modeling.
4. Co-benefit calculation on the hybrid power system.

**Author Contributions:** Methodology, M.J.A.; software, H.A.Z.A.-b. and M.J.A.; formal analysis, H.A.Z.A.-b. and M.J.A.; investigation, H.A.Z.A.-b.; resources, K.N.; data curation, H.A.Z.A.-b.; writing—original draft preparation, H.A.Z.A.-b. and M.J.A.; writing—review and editing, H.A.Z.A.-b., M.J.A., and L.M.M.; visualization, N.N.A.-s.; supervision, M.V.R. and K.Z. All authors have read and agreed to the published version of the manuscript.

**Funding:** This research received no external funding.

**Institutional Review Board Statement:** Not applicable.

**Informed Consent Statement:** Not applicable.

**Data Availability Statement:** The data presented in this study are available on request from the corresponding author.

**Conflicts of Interest:** The authors declare no conflict of interest.

**Nomenclature**

| | |
|---|---|
| *u* | velocity vector in x-axis (m/s) |
| P | pressure (Pa) |
| T | temperature (K) |
| M | gas molecular weight (kg/mol) |
| *x* | mole fraction |
| *y* | mass fraction |
| D | diffusion coefficient (m$^2$/s) |
| C$_p$ | specific heat capacity (J/kg·K) |
| *k* | gas thermal conductivity (W/m·K) |
| $k_p$ | hydraulic permeability (m$^2$) |
| F | Faraday's constant |
| $\dot{q}$ | the generated heat (W/m$^2$) |
| S | specific entropy (J/mol·K) |
| R | universal gas constant (J/mol·K) |

**Subscripts**

| | |
|---|---|
| $i$ | Hydrogen in anode |
| $j$ | Oxygen in cathode |
| $w$ | water |
| $mem$ | membrane |
| $a$ | anode |
| $c$ | cathode |
| $l$ | liquid |

**Greek**

| | |
|---|---|
| $\eta_{act}$ | activation overpotential |
| $\rho$ | density (kg/m$^3$) |
| $\varepsilon$ | porosity |
| $\beta$ | modified convective heat transfer coefficient (W/m$^3$) |
| $\lambda_e$ | electronic conductivity |
| $\mu$ | viscosity (kg/m.s) |
| $\zeta$ | Stoichiometric flow ratio |

**Acronym**

| | |
|---|---|
| PEM | proton-exchange membrane |
| FC | fuel cell |
| PV | photovoltaic |
| COE | cost of energy |
| GDL | gas diffusion layer |
| CL | catalyst layer |
| ACL | anode catalyst layer |
| CCL | cathode catalyst layer |
| MEA | membrane electrode assembly |
| DBCs | Dirichlet's boundary conditions |
| NBCs | Neumann's boundary conditions |
| AGFC | anode gas flow channel |
| CGFC | cathode gas flow channel |

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
