# Peer review of "Design and Simulation Studies of Hybrid Power Systems Based on Photovoltaic, Wind, Electrolyzer, and PEM Fuel Cells"

_energies, doi:10.3390/en14092643_

Round 1

Reviewer 1 Report

There is no new contribution in this work.

Author Response

Reviewer #1

Comment 1.  “There is no new contribution in this work.”

Response: We sincerely thank the respected reviewer for his evaluation of our research paper which prompted us to review our manuscript and try to increase its quality and relevance. In addition to this comment, we have received many valuable comments from other reviewers. We believe that thanks to the comments of the reviewers, we have made many substantial amendments that have greatly improved the quality and relevance of the manuscript.

Reviewer 2 Report

In recent years, the need to reduce environmental impacts and increase flexibility in the energy sector has led to increased penetration of renewable energy sources and the shift from concentrated to decentralized generation. A fuel cell is an instrument that produces electricity by chemical reaction. Fuel cells are a promising technology for ultimate energy conversion and energy generation. Proton exchange membrane fuel cell (PEMFC) which includes a membrane, and two electrodes, has grown up with huge attraction because of its simple operation and fuel availability. The proton exchange membrane fuel cells (PEMFCs) are promising energy devices for stationary and mobile applications because of high power density, high efficiency, low operating temperature, low emissions, low noise, and great environmental compatibility. The PEMFCs are composed of gas diffusion layer (GDL) including gas diffusion backing (GDB) and microporous layer (MPL), membrane electrode assembly (MEA), and bipolar plates with gas channels. The fibrous gas diffusion layer is a core component of a PEMFC, enabling transport of gases, liquids and electricity within the cell. In this paper, a highly reliable hybrid system was developed. This system consists of the following main components: wind turbine, fuel cell, PV, electrolyzer, inverter and tanks. The system works as follows: At low load hours, wind turbines, PV and electrolysis units produce extra power. Although the topic in this work was interesting, the presentation in this manuscript was very poor. This manuscript should be rejected for published in Energies. However, if the authors are willing to make the substantial revisions according to my comments, I would be glad to re-review this manuscript. Here are my detailed comments:

  1. The introduction should be reconstructed to present a coherent literature review. It may help the authors by answering the following questions: Why are these works relevant? Which specific problems were addressed? How are the previous results related with the latest work? What are the outstanding, unresolved, research issues? Answering the questions leads to the novelty of the proposed work naturally.
  2. In Table 3, the authors should give the explanations for the difference of data collected from different sources.
  3. When entering the fuel cell at maximum load at peak hours as energy is produced to meet grid demand. If the tank is full, the energy will convert to a dump load if the PV generation is less than the load demand, then the FCs will produce the energy to cover the rest of the demand when there is no solar radiation, and the wind turbine generates energy continuously. The authors should give some explanation on above conclusions.
  4. Please check all Equations double.
  5. It is suggested to discuss what the main advantages the proposed Mathematical Model of PEM and Numerical Method have.
  6. I am quite interested in some parametric study with the proposed Mathematical Model of PEM and Numerical Method. The manuscript could be more substantial if the authors do so. At least, the authors need to write some statements that how the proposed Mathematical Model of PEM and Numerical Method can be used for the parametric study.
  7. Proton exchange membrane fuel cells have attracted attention from energy devices such as portable, mobile and stationary devices, since it helps effective reductions of energy shortage and environment pollution. Besides the proposed Mathematical Model of PEM and Numerical Method, fractal model is a very important tool, which can be used to investigate proton exchange membrane fuel cells, see [International Journal of Hydrogen Energy, 2018, 43(37):17880-17888; International Journal of Heat and Mass Transfer, 2019, 137:365-371]. Authors should introduce some related knowledge to readers. I think this is essential to keep the interest of the reader.
  8. Although the results look “making sense”, the authors should dig deeper in the results by presenting some in-depth discussion, such as implications of the results, such as possible application of them.
  9. Please, expand the conclusions in relation to the specific goals and the future work.

Author Response

Reviewer #2

Comment 1.  “The introduction should be reconstructed to present a coherent literature review. It may help the authors by answering the following questions: Why are these works relevant? Which specific problems were addressed? How are the previous results related with the latest work? What are the outstanding, unresolved, research issues? Answering the questions leads to the novelty of the proposed work naturally.”

Response: We are grateful for the reviewer’s comment. All directions of the respected reviewer have been taken into consideration regarding the introduction. The introduction has been completely reconstructed (pages 1, 2, 3, and 4 lines (69-163)) based on the answers to the questions referred to in the reviewer's comment as verified on the paper-marked version.

Comment 2.  “In Table 3, the authors should give the explanations for the difference of data collected from different sources”

Response: We are thankful for the reviewer’s comment. We would like to clarify that solar radiation data in Table 3 is a reliable data were obtained from the NASA Atmospheric Data Center. As verified on the paper-marked version, the NASA Atmospheric Data Center website has been included as a reference from which the data for Table 3 were obtained.  Regarding the difference of the data, this is due to the difference in the rates of solar radiation and clearness index for all months of the year as a result of the climate diversity for the studied region.

Comment 3.  “When entering the fuel cell at maximum load at peak hours as energy is produced to meet grid demand. If the tank is full, the energy will convert to a dump load if the PV generation is less than the load demand, then the FCs will produce the energy to cover the rest of the demand when there is no solar radiation, and the wind turbine generates energy continuously. The authors should give some explanation on above conclusions.”

Response: The authors sincerely thank the reviewer for the valuable comment. As we mentioned in (3.7. Operational Control Strategy). We would like to show that the proposed hybrid system was proposed because the studied region is characterized by a moderate climate (hot and dry climate), depending on the data received from NASA, and this is what makes the PV work with high performance. Therefore, the produced power will cover the required load demand throughout the months of the year. Moreover, we have integrated the wind turbine into the proposed system to make it more reliable through the power produced from the wind turbine in times of bad climate to fill the shortage of power produced from PV.

Comment 4.  “Please check all Equations double.”

Response: The authors thank the reviewer for the valuable comment. All the equations have been checking and we corrected typos error in Equation 17 (page 6 line 238) as verified on the paper-marked version.

Comment 5.  “It is suggested to discuss what the main advantages the proposed Mathematical Model of PEM and Numerical Method have.”

Response: The authors sincerely thank the reviewer for this comment. We would like to clarify that the proposed Mathematical Model of PEM and Numerical Method model is used for the purpose of optimizing the performances of the suggested hybrid energy system. Also, it can be used to collect the optimal values of some critical operating parameters which are directly effect on the performance of the hybrid energy system ,in particular the performance of PEMFC.

Comment 6.  “I am quite interested in some parametric study with the proposed Mathematical Model of PEM and Numerical Method. The manuscript could be more substantial if the authors do so. At least, the authors need to write some statements that how the proposed Mathematical Model of PEM and Numerical Method can be used for the parametric study.”

Response: The authors very much appreciate the reviewer for the valuable comment. The proposed Mathematical Model of PEM and Numerical Method can be used for study of the effect of the following parameters:

  1. The effect of the channel geometrical configurations on the PEMFC mechanical behaviour.
  2. The effect of seal (gasket) properties on the contact pressure distribution inside PEMFC.
  3. The effect of GDL porosity and the catalyst layer thickness on the performance of a PEMFC.

These above points are listed immediately after the conclusions ((pages 25 and 26 lines 630–637) in the paper-marked version) as the most important forthcoming work that may be considered further research topics related to our current research.

Comment 7.  “Proton exchange membrane fuel cells have attracted attention from energy devices such as portable, mobile and stationary devices, since it helps effective reductions of energy shortage and environment pollution. Besides the proposed Mathematical Model of PEM and Numerical Method, fractal model is a very important tool, which can be used to investigate proton exchange membrane fuel cells, see [International Journal of Hydrogen Energy, 2018, 43(37):17880-17888; International Journal of Heat and Mass Transfer, 2019, 137:365-371]. Authors should introduce some related knowledge to readers. I think this is essential to keep the interest of the reader.”

Response: The authors thank the reviewer for this comment. The authors' effort focused on the problem of discontinuous electric power to the studied area from the national grid and trying to solve it through a proposed hybrid system has higher reliability to provide continuous generation. Since the results of design and simulation of the studied hybrid power system with the economic analysis to estimate the capital cost of the project in the present work are already quite high, we shall consider the effects of fractal studies on the hybrid power system in our future work. This is because we share the opinion of the respected reviewer that the fractal studies is very important tool, especially in determining the parameters that would improve the performance of the system and thus increase the total amount of power produced from it. We also extend our sincere thanks to the respected reviewer for referring to the two papers [International Journal of Hydrogen Energy, 2018, 43(37):17880-17888 and International Journal of Heat and Mass Transfer, 2019, 137:365-371] as they have been adopted (in the paragraph below) as a literature studies in the introduction (as verified on the paper-marked version (page 3 lines 130–143)) due to their connection with our current study.

                         “An analytical modeling has been carried out in [18] to investigate the influence of many parameters such as the porosity, unit cell aspect ratio, fiber radius, and molar concentration on the transverse permeability of gas diffusion layer (GDL). The fibrous porous media (porosity and fiber radius), the zeta potential, and the electrolyte solution's physical properties have been clearly stated in the suggested modeling. A comparison was made be-tween the results obtained from the proposed modeling and the results of a number of previous literature studies, and it was found that there is a great relative match between these results. To measure the efficient electrolyte diffusivity in porous media while taking into account the influence of electrical double layer (EDL), a fractal modeling of the porous media's fractal theory and capillary model have been suggested in [19]. The proposed modeling specifically addresses the electro-kinetic parameters as well as the porous media's micro-structural parameters. To verify the validity of the results, a comparison was made between the modeling results and data for the results of an experimental study. It was found that there is an acceptable match between these results.”

Comment 8.  “Although the results look “making sense”, the authors should dig deeper in the results by presenting some in-depth discussion, such as implications of the results, such as possible application of them.”

Response: The authors very much appreciate the reviewer for the valuable comment. Although the results and discussion in the manuscript are quit high and include the physical explanation of each result, the comment of the respected reviewer was taken into consideration by reinforcing the discussion of the results by adding other numerical findings, as shown on the paper-marked version.

Comment 9.  “Please, expand the conclusions in relation to the specific goals and the future work.”

Response: We are thankful to the reviewer for the valuable comment. The conclusion has been completely reconstructed (pages 25 and 26 lines (631-659)) depending on the identified research gap re-stating, the formulated research question, and the inclusion of the most important results obtained from the study as verified on the paper-marked version. In addition, we have included a dedicated paragraph immediately after the conclusions ((page 26 lines 660–668) in the paper-marked version) that show the most important forthcoming work that may be considered further research topics related to our current research.

Reviewer 3 Report

The article "Design and Simulation Studies of Hybrid Power Systems Based on Photovoltaic, Wind, Electrolyser and PEM Fuel Cells" is of interest for Energies' readers and deals with a hot topic.

Authors should stress the novelty compared to a simple HOMER simulation. This is crucial and not very clear even if a lot of work done by the authors is evident.

Literature is poor. I suggest check recent articles in top-Journals such as Renewable Energy, Renewable & Sustainable Energy Reviews, Journal of Electrochemical Energy Conversion and Storage, etc.

An example for each source is https://doi.org/10.1016/j.renene.2020.10.055 https://doi.org/10.1016/j.rser.2020.109777 https://doi.org/10.1115/1.4041864 

The load profile is almost the same during the year. Please give further support to this assumption.

Please harmonize tables and graphs following the font of the template.

Enrich the discussions of results comparing with other literature and justify differences. See other high-ranked sources such as International Journal of Hydrogen Energy, Energy Conversion and Management and others. Examples can be found in https://doi.org/10.1016/j.ijhydene.2020.07.251 https://doi.org/10.1016/j.enconman.2021.113993 and others

Conclusions must be extended re-stating the identified research gap, the research question formulated by the authors, the potential and limitations of their outcomes and what differs this study from a simple case study.

Author Response

Reviewer #3

Comment 1.  “Authors should stress the novelty compared to a simple HOMER simulation. This is crucial and not very clear even if a lot of work done by the authors is evident.”

Response: We are grateful for the reviewer’s comment. We would like to clarify that, in addition to the HOMER simulation, we proposed a mathematical model of PEMFC as well as numerical method simulation which is used for the purpose of optimizing the performances of the suggested hybrid energy system. Also, it can be used to collect the optimal values of some critical operating parameters which are directly effect on the performance of the hybrid energy system ,in particular the performance of PEMFC.

Comment 2.  “Literature is poor. I suggest check recent articles in top-Journals such as Renewable Energy, Renewable & Sustainable Energy Reviews, Journal of Electrochemical Energy Conversion and Storage, etc. An example for each source is https://doi.org/10.1016/j.renene.2020.10.055 https://doi.org/10.1016/j.rser.2020.109777 https://doi.org/10.1115/1.4041864

Response: The authors sincerely thank the reviewer for this comment. This suggestion has taken into consideration where the introduction has been completely reconstructed (pages 1, 2, 3, and 4 lines (69-163)). Literature have been enriched by taking an appropriate number of related articles in top-Journals within the introduction as verified on the paper-marked version.

Comment 3.  “The load profile is almost the same during the year. Please give further support to this assumption.”

Response: The authors thank the reviewer for the valuable comment. We would like to clarify that the Bahr AL-Najaf region load was already set at 12000 kWh/day. The maximum load on a seasonal scale is assumed to be 1.5MW. It must be observed that the yearly maximum load of 1.5MW happens in June. The peak season (between June and July) has the highest demand, while the low season (between December and January) has the lowest demand. The following an independent paragraph is dedicated to this information in (3.1. Case Study) section as verified on the paper-marked version (page 12 lines (378-382)).

                   “The Bahr AL-Najaf region load was already set at 12000 kWh/day. The maximum load on a seasonal scale is assumed to be 1.5MW. It must be observed that the yearly maximum load of 1.5MW happens in June. The peak season (between June and July) has the highest demand, while the low season (between December and January) has the lowest demand.”

Comment 4.  “Please harmonize tables and graphs following the font of the template.”

Response: We sincerely thank the reviewer for his comment. The comment has been taken into consideration and we harmonized tables and graphs following the font of the template, as verified on the paper-marked version.

Comment 5.  “Enrich the discussions of results comparing with other literature and justify differences. See other high-ranked sources such as International Journal of Hydrogen Energy, Energy Conversion and Management and others. Examples can be found in

 https://doi.org/10.1016/j.ijhydene.2020.07.251

https://doi.org/10.1016/j.enconman.2021.113993 and others”

Response: We thank the respected reviewer for this comment. Although the results and discussions in the manuscript are quit high and comprehensive the physical explanation of each result, the comment of the respected reviewer was taken into consideration by reinforcing the discussion of the results by adding other numerical findings, as shown on the paper-marked version.

Comment 6.  “Conclusions must be extended re-stating the identified research gap, the research question formulated by the authors, the potential and limitations of their outcomes and what differs this study from a simple case study.”

Response: We are thankful to the reviewer for the valuable comment. The conclusion has been completely reconstructed (pages 25 and 26 lines (631-659)) depending on the identified research gap re-stating, the formulated research question, and the inclusion of the most important results obtained from the study as verified on the paper-marked version. In addition, we have included a dedicated paragraph immediately after the conclusions ((page 26 lines 660–668) in the paper-marked version) that show the most important forthcoming work that may be considered further research topics related to our current research.

Round 2

Reviewer 2 Report

Very good. It is ok.

Author Response

Thanks for the referee time in reading our paper and accepting our revised paper 

as per referee suggestions we corected errors

Reviewer 3 Report

The new version addresses the reviewer's concerns in a satisfactory way.

Author Response

Thanks for the referee for accepting our paper and

in the revised MS we corrected few errors